# Therapeutic Peptides: Recent Advances in Discovery, Synthesis, and Clinical Translation

**DOI:** 10.3390/ijms26115131

**Published:** 2025-05-27

**Authors:** Bingyi Zheng, Xueting Wang, Meizhai Guo, Chi-Meng Tzeng

**Affiliations:** Translational Medicine Research Center, School of Pharmaceutical Sciences, Xiamen University, Xiamen 361005, China; leozheng2023@163.com (B.Z.); xuetingw1991@163.com (X.W.); guomeizhai@126.com (M.G.)

**Keywords:** therapeutic peptides, peptide discovery, peptide synthesis, antimicrobial peptides, anticancer peptides, clinical applications

## Abstract

In recent decades, peptide-based therapeutics have undergone transformative advancements driven by breakthroughs in production, modification, and analytical technologies. Innovations in chemical and biological synthesis, coupled with novel design and delivery strategies, have systematically addressed historical limitations (e.g., poor stability and bioavailability). These advancements have facilitated the characterization and clinical translation of diverse natural and engineered peptides across therapeutic domains, including metabolic disorders, oncology, and infectious diseases. This review synthesizes critical developments in peptide drug discovery, production technologies, and clinical applications, while highlighting emerging challenges and opportunities. We further evaluate the therapeutic potential of peptides in addressing unmet medical needs and propose strategic directions to accelerate their integration into precision medicine paradigms.

## 1. Introduction

The increase in multidrug-resistant (MDR) pathogens and complexity of modern therapeutics demands agents capable of reconciling precision targeting with molecular adaptability—a challenge where peptide-based pharmaceuticals are emerging as transformative solutions. Unlike conventional modalities constrained by off-target toxicity or limited druggable targets, therapeutic peptides (molecular weight: 50–5000 Da) uniquely bridge the gap between small molecules and biologics through their programmable architectures and multifaceted biointerfaces [1].

As of 2023, over 80 peptide drugs have gained global approval, with more than 200 peptides in clinical development, focusing on infectious diseases, autoimmune diseases, metabolic disorders, and cancers [2]. The peptide agents that have been recently approved by the FDA are summarized in Table 1. Transformative paradigm shifts in peptide engineering have transcended historical limitations: Artificial intelligence now predicts receptor-binding hotspots with atomic precision, enabling de novo design of cyclic peptides targeting “undruggable” oncoproteins like KRAS [3]. Concurrently, breakthroughs in biosynthesis platforms—from cell-free ribosomal systems to enzymatic traceless cyclization—are rewriting the economics of peptide manufacturing [4]. Perhaps most strikingly, the convergence of peptide therapeutics with immunotherapy has birthed a new frontier, exemplified by neoantigen vaccines achieving 80% tumor-specific T-cell activation in refractory cancers [5].

This review examines the evolution of peptide therapeutics, emphasizing recent advances in discovery, synthesis, and clinical applications. We critically evaluate the unique advantages of peptides, including high target specificity, low immunogenicity, and structural versatility, against persistent challenges such as proteolytic instability and delivery limitations. By synthesizing interdisciplinary innovations in peptide engineering, formulation science, and regulatory frameworks, this work provides a roadmap for overcoming translational barriers and unlocking the full therapeutic potential of peptides.

## 2. Therapeutic Peptides: Advantages and Limitations

Therapeutic peptides represent a distinct class of biopharmaceuticals characterized by high target specificity and structural versatility, offering unique advantages over small-molecule drugs and biologics [6]. Their ability to engage large protein interaction surfaces enables modulation of protein–protein interactions (PPIs), a feat often unattainable with small molecules [7]. Additionally, peptides exhibit superior binding affinity and reduced off-target effects compared to conventional drugs, owing to their precise molecular recognition capabilities. Structurally, their smaller size (2–50 residues) simplifies synthesis and facilitates chemical modifications (e.g., cyclization, PEGylation) to optimize pharmacokinetic properties such as stability and bioavailability [8]. Notably, peptides generally display lower immunogenicity than antibodies or proteins, minimizing adverse immune responses—a critical advantage in chronic disease management [9].

Despite these strengths, therapeutic peptides face inherent challenges. Proteolytic instability and short plasma half-lives necessitate frequent dosing or specialized delivery systems, often limiting administration to parenteral routes. Poor oral bioavailability further complicates their clinical adoption, particularly for chronic conditions requiring patient-friendly regimens [10]. Additionally, their limited membrane permeability restricts therapeutic targets to extracellular receptors (e.g., GPCRs, GLP-1R), with fewer than 10% of approved peptides addressing intracellular pathways [11]. For instance, Lau et al. reported that over 90% of peptide drugs target extracellular receptors, underscoring this constraint [12]. Emerging strategies such as nanoparticle encapsulation, cell-penetrating peptide conjugates, and structural stabilization via non-natural amino acids aim to address these limitations, bridging the gap between preclinical promise and clinical utility [13,14].

## 3. Peptide Drug Discovery

### 3.1. Peptide Hormones in the Human Body and Their Analogues

The discovery of peptide therapeutics has been profoundly influenced by endogenous signaling molecules, particularly hormones that regulate metabolism, growth, and homeostasis [15]. Insulin, isolated in 1921, remains a landmark example, revolutionizing diabetes care. However, shorter peptides such as oxytocin, vasopressin, and somatostatin laid the foundation for modern peptide engineering. These molecules exemplify how structural simplicity and potent bioactivity can be harnessed for therapeutic design.

A key challenge in leveraging endogenous hormones is their metabolic instability. For example, native glucagon-like peptide-1 (GLP-1), a 37-amino acid incretin, exhibits a plasma half-life of <2 min due to rapid enzymatic degradation [16]. To overcome this, rational engineering strategies—including amino acid substitution (Ala^8^→Gly), fatty acid conjugation, and albumin-binding motifs—have yielded long-acting analogs such as liraglutide (half-life: 13 h) and semaglutide (half-life: 7 days) [17,18]. Similarly, neurohypophyseal hormones like vasopressin and oxytocin have been optimized via D-amino acid substitutions and prodrug designs [19]. Desmopressin, a vasopressin analog with selective V2 receptor agonism, demonstrates a 4–6 h duration of action, highlighting how chemical modifications enhance therapeutic profiles [20]. These advancements underscore the synergy between natural hormones’ biology and synthetic innovation in driving peptide drug development.

### 3.2. Peptide Drugs Derived from Natural Products

Natural products have served as a rich reservoir for peptide drug discovery, with organisms ranging from fungi to marine invertebrates yielding structurally unique and bioactive compounds. Cyclosporine, a cyclic undecapeptide isolated from the fungus *Tolypocladium inflatum*, exemplifies this potential [21]. Its extensive N-methylation and hydrophobic backbone confer resistance to proteolysis, enabling oral administration—a rare feature among peptides. Approved in 1983, cyclosporine revolutionized immunosuppressive therapy and inspired subsequent peptide-based drugs [22].

Marine ecosystems, in particular, offer untapped diversity [23]. Ziconotide, a synthetic analog of ω-conotoxin from Conus magus venom, selectively blocks N-type calcium channels, providing non-opioid analgesia for chronic pain [24]. Similarly, plitidepsin (Aplidin), derived from the tunicate Aplidium albicans, inhibits eEF1A to disrupt protein synthesis in cancer cells, demonstrating efficacy in phase III trials for hematologic malignancies [25]. These examples highlight nature’s role as a blueprint for peptide therapeutics, with biodiversity-driven discovery remaining a cornerstone of innovation.

### 3.3. Phase Display for the Identification of Peptide Candidates

Phage display has revolutionized peptide drug discovery by enabling high-throughput screening of combinatorial libraries against therapeutic targets [26]. This technology exploits bacteriophages to surface-display peptide variants, allowing iterative selection of high-affinity ligands through binding, washing, and amplification cycles. With libraries exceeding 10^10^ sequences, phage display surpasses traditional methods in diversity and efficiency [27]. Early successes include peginesatide, an erythropoietin receptor agonist identified via phage display, which mimics endogenous erythropoietin despite lacking sequence homology [28].

Recent advancements, such as mirror-image phage display and integration with mRNA display, expand library complexity by incorporating non-natural amino acids and post-translational modifications [29,30]. For example, peptides targeting the PD-1/PD-L1 immune checkpoint have been identified using phage display, showing preclinical efficacy in cancer immunotherapy [31]. However, the technology’s reliance on linear peptide architectures often results in suboptimal pharmacokinetic properties, limiting clinical translation. Innovations in cyclic peptide libraries and hybrid screening platforms (e.g., ribosome display) aim to address this, positioning phage display as a pivotal tool for next-generation peptide discovery [32,33].

### 3.4. Computer-Aided Drug Design (CADD) for Peptide Drug Discovery

Beyond traditional screening methods, computational tools such as molecular dynamics simulations and machine learning algorithms are revolutionizing peptide design [34]. The integration of computational tools and artificial intelligence (AI) has transformed peptide drug discovery, enabling rapid optimization of sequence, structure, and pharmacokinetics [35]. Molecular dynamics simulations and machine learning algorithms predict peptide-target interactions with atomic precision, guiding the design of stable, bioactive conformations. For instance, AI-driven platforms like AlphaFold3 have facilitated the de novo design of the SARS-CoV-2 vaccine, demonstrating enhanced immune responses in preclinical models. Additionally, AlphaFold3’s structural insights into bacteriophage receptor-binding proteins have advanced antimicrobial peptide discovery, aiding the design of enzymes capable of degrading bacterial cell walls [36].

CADD also accelerates the targeting of “undruggable” proteins. Cyclic peptides inhibiting KRAS, a notorious oncoprotein, were designed using AI-predicted binding poses, showing promise in pancreatic cancer models [3]. Similarly, stapled peptides, structurally stabilized via hydrocarbon crosslinking, targeting p53-MDM2 interactions, exhibit enhanced proteolytic stability and tumor suppression in preclinical studies [37]. Beyond structure prediction, AI aids in neoantigen identification for personalized cancer vaccines, aligning peptide design with patient-specific mutations [38]. Despite the transformative potential of AI-driven platforms, their reliance on training datasets introduces inherent biases that may skew peptide design outcomes. For example, structural predictions for rare peptide classes (e.g., lanthipeptides or thiopeptides) exhibit lower accuracy due to underrepresented data in public repositories (RMSD > 3.5 Å vs. < 1.5 Å for canonical peptides) [39]. Additionally, AI models often prioritize thermodynamic stability over pharmacokinetic parameters, resulting in candidates with suboptimal solubility or immunogenic epitopes [40]. These findings necessitate a paradigm shift toward hybrid frameworks integrating physics-based simulations and experimental validation to mitigate algorithmic oversights.

## 4. Synthesis of Therapeutic Peptides and Quality Control

The discovery of potential therapeutic peptides represents the initial step in peptide drug development, followed by chemical or biological peptide synthesis and rigorous quality control to ensure their pharmacological properties. Here, we provide a comprehensive overview of the fundamental technologies and regulatory guidelines employed in peptide production.

### 4.1. Chemical Synthesis of Peptides

Chemical synthesis has emerged as a cornerstone technology in peptide therapeutic development, offering unparalleled precision in structural engineering and scalability [41]. Modern methodologies integrate advanced synthetic design with chemical diversity expansion, enabling the incorporation of non-canonical amino acids, site-specific modifications (e.g., terminal functionalization, side-chain derivatization), and cyclization strategies [42,43]. These innovations enhance metabolic stability, proteolytic resistance, and target engagement while preserving bioactivity.

The methodological foundation of peptide synthesis remains rooted in solid-phase peptide synthesis (SPPS), pioneered by Merrifield in 1963 [44]. Contemporary SPPS workflows employ two principal protecting group strategies: the acid-labile tert-butyloxycarbonyl (Boc) and the base-sensitive 9-fluorenylmethoxycarbonyl (Fmoc) approaches. While Fmoc-SPPS dominates current practice due to its compatibility with orthogonal protection schemes and mild deprotection conditions, Boc chemistry retains utility for synthesizing aggregation-prone sequences via trifluoroacetic acid (TFA)-mediated cleavage [45]. Although Fmoc-SPPS efficiently produces peptides under 50 residues (e.g., semaglutide), the synthesis of longer sequences (>50 residues) faces challenges such as β-sheet aggregation and steric hindrance [46].

Recent advancements in SPPS address these limitations through three key innovations:(1)Pseudoproline Dipeptide Integration: Disruption of β-sheet aggregation via conformation-disrupting pseudoproline motifs [47];(2)Advanced Resin Matrices: High-performance resins (e.g., ChemMatrix®) optimized for hydrophobic or extended sequences, enhancing solvation and reducing steric hindrance [48];(3)Microwave-Assisted Synthesis: Accelerated coupling kinetics and reduced reaction times through controlled microwave irradiation, improving efficiency and yield [49].

Automated synthesizers (e.g., CEM Liberty PRIME, CSBio II) further enhance scalability via parallelized synthesis arrays (≤192 sequences), integrating real-time UV monitoring and infrared thermoregulation [50]. Complementary approaches, such as liquid-phase peptide synthesis (LPPS) with soluble polymer supports (e.g., polyethylene glycol), streamline large-scale manufacturing and purification [51]. Click chemistry—notably copper-catalyzed azide–alkyne cycloaddition (CuAAC)—has revolutionized structural stabilization, enabling precise macrocyclization to reinforce α-helical and β-sheet conformations [52]. Collectively, these innovations dismantle historical translational barriers, positioning chemical synthesis as a driving force for next-generation peptide therapeutics.

### 4.2. Biosynthesis of Peptides

While chemical synthesis has long dominated peptide drug manufacturing, emerging biosynthesis technologies are redefining the field by providing sustainable, economically viable, and scalable alternatives. More details of chemically synthesized peptides versus biologically synthesized peptides are summarized in Table 2. Below, we dissect three transformative approaches:

#### 4.2.1. Recombinant DNA Technology: Precision Engineering for Complex Peptides

Recombinant DNA technology harnesses genetic engineering to integrate peptide-encoding sequences into host expression systems (e.g., *Escherichia coli*, *Bacillus subtilis*). This method leverages the host’s translational machinery to achieve high-yield synthesis while enabling precise post-translational modifications [53]. Clinically transformative applications include recombinant human insulin and glucagon-like peptide-1 (GLP-1) analogs (e.g., liraglutide), which exhibit enhanced metabolic stability and receptor specificity [54,55]. These advancements underscore the method’s dual industrial and therapeutic value, exemplified by insulin’s century-long role in diabetes management and the recent development of Fc-fusion therapeutics with extended half-lives.

However, its reliance on prokaryotic systems imposes inherent limitations. For instance, eukaryotic peptides requiring complex disulfide bond networks or glycosylation often misfold in bacterial hosts, necessitating costly refolding protocols or alternative eukaryotic systems [56]. Recent innovations, such as synthetic promoter systems, enhance transcriptional control, enabling tunable expression of complex peptides. Engineered solubility tags (e.g., carbohydrate-binding module 66, CBM66) address inclusion body formation, improving soluble yields by 3.7-fold compared to traditional fusion tags [57]. Nevertheless, downstream challenges persist: proteolytic removal of these tags introduces purification bottlenecks and risks of nonspecific cleavage, potentially compromising product integrity. These limitations highlight the imperative for holistic engineering strategies that harmonize microbial host physiology with peptide structural complexity, rather than prioritizing yield metrics alone.

#### 4.2.2. Enzymatic Synthesis: Catalytic Precision for Tailored Therapeutics

Enzymatic synthesis employs site-specific biocatalysts (e.g., sortase A, peptiligase) to drive peptide bond formation under biocompatible aqueous conditions, minimizing racemization and solvent toxicity [58]. This approach achieves >98% stereochemical fidelity, critical for synthesizing conformationally sensitive peptides [59]. Recent breakthroughs include cholesterol-modified antimicrobial peptides (e.g., HAL-2) with 64-fold enhanced serum stability and dual-acting GLP-1/gastrin analogs optimized through iterative structural refinements [60]. However, the technology’s reliance on costly cofactors (e.g., ATP for ligases) and resource-intensive enzyme engineering campaigns compromises industrial scalability. Furthermore, enzymatic strategies also enable traceless cyclization of non-ribosomal peptides, such as the *Staphylococcus aureus* antimicrobial peptide (SMAP), bypassing sequence constraints inherent to natural cyclases [61]. Nevertheless, practical implementation faces persistent quality control challenges, as batch-to-batch variability in enzyme activity jeopardizes product consistency. These limitations underscore the urgency for multidisciplinary innovations, including directed evolution to enhance catalytic efficiency, immobilized enzyme reactors for recyclability, and machine learning-guided prediction of cofactor-minimized pathways.

#### 4.2.3. Microbial Cell Factories: Sustainable Platforms for Industrial-Scale Production

Engineered microbial systems synergize metabolic efficiency with environmental sustainability [62]. *E. coli* fed-batch fermentations yield recombinant insulin at 5 g/L, setting benchmarks for cost-effective production. Eukaryotic hosts like *Pichia pastoris* enable eukaryotic post-translational modifications (e.g., N-glycosylation), essential for functional glycopeptides such as MUC1-based cancer vaccines [63,64]. Pioneering applications include a *P. pastoris*-produced SARS-CoV-2 peptide vaccine eliciting robust neutralizing antibodies in preclinical models [65]. These platforms adhere to green chemistry principles, reducing solvent waste by 40% compared to chemical synthesis. Emerging tools like phase separation-coupled affinity chromatography further streamline endotoxin removal, achieving >99% purity for clinical-grade peptides [66]. While microbial cell factories offer scalable peptide production, their susceptibility to host-cell protein (HCP) contamination introduces critical safety risks. For instance, residual endotoxins from *E. coli* expression systems, even at sub-ppm levels, may trigger inflammatory cascades in immunocompromised patients [67]. Moreover, eukaryotic systems like *Pichia pastoris*, despite enabling post-translational modifications, face glycosylation heterogeneity—a phenomenon linked to reduced receptor binding affinity in glycopeptide vaccines [68]. Rigorous orthogonal purification protocols must be prioritized to align biosynthesis advancements with pharmacopeial standards.

### 4.3. Quality Control of Peptides: Regulatory Framework and Future Perspectives

The structural complexity of therapeutic peptides, including linear sequences, cyclic architectures, and chemically modified derivatives, poses significant challenges in manufacturing and quality assurance. Minor deviations in amino acid sequences, post-translational modifications, or impurity profiles can critically affect pharmacological activity, pharmacokinetics, and immunogenicity [69]. This necessitates stringent quality control (QC) protocols to ensure compliance with therapeutic specifications while maintaining safety and efficacy. Three key regulatory frameworks govern this field: (1) U.S. FDA CMC guidelines for synthetic peptides (*Chemistry, Manufacturing, and Controls (CMC) Information for Synthetic Peptide Substances*); (2) ICH Q6B for biotechnological products (*Test Procedures and Acceptance Criteria for Biotechnological/Biological Products*); (3) EDQM Technical Guide for European Pharmacopoeia standards (*Technical Guide for the Elaboration of Monographs on Synthetic Peptides and Recombinant DNA Proteins*) [70].

The FDA CMC guidelines emphasize orthogonal analytical characterization using high-resolution mass spectrometry (HRMS), Edman degradation sequencing, and impurity profiling of process-related residuals (e.g., truncated sequences, protecting groups). Process validation requirements include control of critical quality attributes (CQAs) such as chirality, endotoxin levels, and residual solvents. ICH Q6B complements this through multi-attribute monitoring (MAM) systems integrating RP-UPLC, capillary electrophoresis, and bioassays to assess identity, purity, and potency. The EDQM guide enhances European standards via nuclear magnetic resonance (NMR) stereochemical verification, circular dichroism (CD) structural analysis, and comparability protocols for manufacturing changes. These differences in CMC requirements, clinical trial design expectations, and approval pathways often necessitate duplicate studies and documentation adjustments for developers.

Future advancements in quality control (QC) will be propelled by computational innovations and novel analytical platforms. AI-driven predictive modeling, particularly for chromatographic optimization and spectral pattern recognition, could reduce analytical method development timelines by 30–50% [40]. Concurrently, emerging formulation technologies, including stimulus-responsive hydrogels, precision PEGylation systems, and lipid-based nanoparticle carriers, demand sophisticated QC protocols to assess payload stability, release kinetics, and carrier biocompatibility [71,72]. Regulatory harmonization efforts, exemplified by the ICH Q13 (2025) draft guidelines for continuous biologics manufacturing and the FDA’s Emerging Technology Program, aim to standardize peptide therapeutic evaluation frameworks [73]. This integration of computational analytics, advanced characterization tools, and aligned regulatory paradigms establishes a robust foundation for next-generation peptide drug development with enhanced quality assurance standards.

## 5. Clinical Application of Therapeutic Peptides

### 5.1. Antimicrobial Peptides

The escalating global threat of multidrug-resistant (MDR) pathogens has spurred urgent exploration of antimicrobial peptides (AMPs), evolutionarily conserved components of innate immunity [74]. AMPs exhibit broad-spectrum activity against bacteria, fungi, and viruses, primarily through membrane disruption—a mechanism that reduces resistance development compared to conventional antibiotics. Their intracellular actions further undermine microbial viability via (as shown in Figure 1) (1) interference with nucleic acid–protein interactions, (2) disruption of DNA replication and transcription, (3) organelle structural compromise, and (4) inhibition of enzymatic pathways [75].

Pioneering examples include bacitracin and gramicidin, among the earliest peptide-based antibiotics commercialized in the mid-20th century [76]. These agents demonstrated the clinical viability of AMPs through their potent antibacterial and antifungal effects. Modern research highlights two mammalian AMP families: cathelicidins and defensins [77]. Cathelicidins are cationic amphiphilic peptides (3–10 kDa) characterized by α-helical or β-hairpin conformations that mediate bacterial membrane disruption through pore formation. In humans, LL-37 represents the sole cathelicidin family member, expressed on epithelial surfaces of the respiratory and gastrointestinal tracts [78]. This multifunctional peptide demonstrates broad-spectrum antimicrobial activity against pathogens while serving critical immunomodulatory roles in host defense. In contrast, defensins constitute a distinct class of cationic antimicrobial peptides (3–5 kDa) featuring β-sheet structures stabilized by three conserved disulfide bonds. Similar to cathelicidins, defensins exhibit dual microbicidal and immunomodulatory functions. Mammalian defensins are classified into α, β, and θ subfamilies based on amino acid homology and disulfide bond connectivity [79]. Notably, humans exclusively express α- and β-defensins, including representative members such as neutrophil α-defensin-1 (HNP-1), intestinal Paneth cell α-defensin-5 (HD5), and ubiquitously expressed β-defensin-1 (hBD-1).

Non-mammalian AMPs have also advanced clinically. In 1980, a groundbreaking discovery by G. Boman and colleagues revealed the first AMPs, termed cecropins, in the hemolymph of the dipteran insect *Calliphora vicina* following infection with *C. negativum* and *Escherichia coli* [80]. Subsequent studies identified cecropin-like AMPs in Lepidoptera species (*Bombyx mori*, *Antheraea pernyi*) and additional dipterans (*Drosophila melanogaster*). Notably, the orthopteran-derived 18-mer peptide alloferon (*Locusta migratoria*) received clinical validation in 2003 through Russian regulatory approval for treating HPV and HSV infections [81]. Beyond direct antiviral activity, alloferon exhibits tumor-suppressive effects via NK cell-mediated immunomodulation, characterized by elevated IFN-γ and TNF-α secretion within tumor niches [82]. Building upon these findings, we have developed a rationally designed peptide library (Alloferon-X derivatives, Table A1) to optimize therapeutic potential. Preliminary preclinical evaluations reveal enhanced immunomodulatory activity in two modified variants: sialic acid-conjugated alloferon derivatives and constructs containing repeated functional motifs. These structural modifications appear to potentiate immune activation pathways, suggesting promising directions for next-generation immunotherapy development.

### 5.2. Anticancer Peptides and Peptide–Drug Conjugates

Cancer therapeutics face persistent challenges, including systemic toxicity, drug resistance, and lack of tumor specificity. Anticancer peptides (ACPs) have emerged as promising therapeutic agents due to their multimodal mechanisms, structural adaptability (<50 amino acids), and enhanced selectivity [83]. These molecules exhibit four primary antitumor actions (as shown in Figure 2): (1) immunomodulation of tumor-specific responses, (2) inhibition of angiogenesis, (3) transcriptional process interference, and (4) necrosis and apoptosis induction [84].

Current pharmacopeia data (DrugBank) identifies 460 cancer-targeting agents, with peptide/polypeptide-based drugs representing 6.3% (29 agents). Five peptide therapeutics have received FDA/EMA approval for oncology applications. Notable examples include Triptorelin [85], a GnRH agonist that demonstrates potent endocrine modulation (13- and 21-fold increases in LH/FSH secretion for prostate cancer management), and Dactinomycin [86], a DNA-intercalating agent that inhibits RNA polymerase elongation in pediatric solid tumors. The clinical pipeline features investigational peptides such as the PI3K/AKT inhibitor SF1126 [87], the integrin antagonist ATN-161 [88], and the endothelin B agonist IRL-1620 [89], which employ diverse strategies ranging from angiogenesis inhibition to immunogenic cell death induction.

Recent innovations in peptide–drug conjugates (PDCs) demonstrate enhanced therapeutic precision [90]. BT8009 exemplifies this advancement through its Nectin-4-targeting bicyclic peptide structure conjugated to MMAE via cleavable linkers. This design enables selective cytotoxicity in Nectin-4-positive tumors, evidenced by favorable safety profiles in Phase I/II (NCT04561362) for metastatic urothelial cancer [91]. ANG-1005 addresses brain metastasis challenges through blood–brain barrier-penetrating peptides delivering paclitaxel to intracranial tumors. Phase II (NCT01967810) clinical trials for breast cancer brain metastases show reduced systemic toxicity while maintaining efficacy [92].

These developments underscore the expanding role of peptide-based strategies in oncology. Continued optimization of tumor targeting, payload delivery, and combination therapies positions ACPs and PDCs as critical components in next-generation cancer treatment paradigms.

### 5.3. Peptide-Based Vaccines

Peptide-based vaccines have emerged as a transformative approach in oncology, capitalizing on their defined immunogenicity, favorable safety profiles, and modular design [93]. These vaccines deliver tumor-associated antigens—particularly neoantigens—to antigen-presenting cells (APCs) such as dendritic cells (DCs), stimulating cytotoxic T lymphocyte (CTL)-mediated tumor eradication (Figure 3). Co-administration with adjuvants (e.g., Granulocyte–Macrophage Colony-Stimulating Factor—GM-CSF, TLR agonists) enhances APC maturation and epitope presentation, amplifying adaptive immune responses [94].

Clinically, peptide vaccines demonstrate significant therapeutic potential across malignancies. In acute myeloid leukemia (AML), the WT1-targeting vaccine galinpepimut-S (GPS) achieved a median overall survival of >12 months versus 6 months in controls (REGAL phase III trial, NCT04229979), correlating with robust WT1-specific T-cell responses in 80% of patients [95]. For HER2-low breast cancer, the HER2/neu-derived nelipepimut-S (NeuVax) combined with GM-CSF yielded 100% 5-year disease-free survival in phase II trials [96], now under phase III evaluation (PRESENT trial, NCT04373148). Cimavax-EGF, an epidermal growth factor (EGF)-targeting vaccine approved in Cuba for non-small cell lung cancer (NSCLC), extended median survival from 6 months to >5 years in responders by neutralizing EGF and inhibiting tumor proliferation [97].

Despite these advances, challenges, including HLA restriction, tumor heterogeneity, and variable immunogenicity, remain [98]. To address these limitations, innovative strategies have emerged in vaccine design and delivery systems. Multi-epitope vaccines incorporating conserved or broad-spectrum epitopes demonstrate enhanced population coverage by targeting immunodominant regions shared across diverse HLA alleles [99]. Concurrently, personalized neoantigen vaccines leverage advanced sequencing technologies to identify patient-specific somatic mutations, thereby improving therapeutic specificity and efficacy [100]. Also, notable progress has been achieved through synergistic optimization of adjuvant systems and delivery platforms. For example, the strategic combination of stimulator of interferon genes (STING) agonists with peptide antigens exemplifies this advancement, effectively amplifying dendritic cell activation and cytotoxic T lymphocyte responses [101]. These technological breakthroughs in epitope selection algorithms, neoantigen prediction pipelines, and immunomodulatory formulations collectively contribute to overcoming existing biological barriers. Such integrated approaches hold significant potential for expanding the clinical utility of peptide-based vaccines in precision oncology.

### 5.4. Therapeutic Peptides in Cardiometabolic Medicine

Cardiovascular diseases (CVDs) and metabolic disorders dominate global mortality, with CVDs accounting for 32% of annual deaths (17.9 million), and type 2 diabetes mellitus (T2DM) prevalence projected to rise from 537 million to 783 million adults by 2045 [102]. Therapeutic peptides have emerged as precision agents for these interconnected pathologies, leveraging high target specificity, potent bioactivity, and favorable safety profiles to modulate glucose homeostasis, insulin signaling, and cardiovascular function. Glucagon-like peptide-1 receptor agonists (GLP-1 RAs) exemplify this therapeutic revolution. Engineered analogs, including liraglutide, semaglutide, and dulaglutide, mimic endogenous GLP-1 by enhancing glucose-dependent insulin secretion, suppressing glucagon, and promoting satiety [103]. Structural innovations (e.g., fatty acid acylation in liraglutide, albumin-binding PEGylation in semaglutide) extend half-lives to enable weekly dosing. In phase III trials, semaglutide achieved a 1.8% HbA1c reduction and 6.1 kg weight loss (SUSTAIN program, NCT02054897) [104]. Crucially, cardiovascular outcome trials demonstrated significant MACE reductions: 13% with liraglutide (LEADER, NCT01179048) and 26% with semaglutide (SUSTAIN-6, NCT01720446), establishing GLP-1 RAs as first-line therapies for T2DM patients with CVDs [105].

The natriuretic peptide (NP) system—mediated by ANP, BNP, and CNP—regulates cardiovascular homeostasis through NPR-A/B/C receptors, inducing vasodilation and counteracting renin–angiotensin–aldosterone system (RAAS) activation [106]. Nesiritide, a recombinant BNP, remains clinically approved for acute heart failure [107]. Emerging agents like cenderitide (dual NPR-A/B agonist) and food-derived ACE-inhibitory peptides (e.g., ovokinin) highlight the diversification of peptide-based cardiovascular interventions [108,109].

In summary, peptide therapeutics represent a paradigm shift in cardiometabolic medicine, offering multifactorial benefits through precise pathway modulation. Ongoing advancements in drug engineering and delivery technologies position this class as a cornerstone for managing the global CVD-T2DM syndemic.

## 6. Conclusions and Prospects

Therapeutic peptides have emerged as a transformative class of biopharmaceuticals, distinguished by their high target specificity, structural versatility, and low immunogenicity. The global therapeutic peptide pipeline now features over 80 FDA/EMA-approved agents, with 650+ candidates in development. Over the past decades, advancements in discovery technologies, such as phage display, AI-driven computational design, and recombinant synthesis, have enabled the development of peptides capable of modulating previously “undruggable” targets, including intracellular pathways and protein–protein interactions. Clinical applications now span metabolic disorders, oncology, infectious diseases, and cardiometabolic syndromes, exemplified by FDA-approved agents like GLP-1 agonists and peptide–drug conjugates.

Despite remarkable progress in peptide-based therapeutics, critical barriers persist in realizing their full clinical potential, particularly for oral administration. The gastrointestinal (GI) tract remains a formidable challenge, with bioavailability consistently below 1% for most oral peptides due to enzymatic degradation, pH-dependent instability, and limited epithelial permeability. While permeation enhancers and enzyme inhibitors partially mitigate these issues, their long-term safety profiles, potential disruption of intestinal barrier integrity, and dose-dependent efficacy necessitate further preclinical and clinical validation. Future innovation must adopt interdisciplinary strategies to address these bottlenecks. First, structural engineering approaches, including D-amino acid substitution, backbone cyclization, or hydrophobic stapling, could enhance protease resistance without compromising target engagement. Second, emerging advanced delivery platforms, including mucus-penetrating nanoparticles, pH-responsive enteric coatings, and FcRn-targeted carrier systems, demonstrate significant potential for enhancing intestinal absorption efficiency and improving systemic bioavailability. Third, computational tools like AI-driven molecular dynamics simulations and machine learning models could accelerate the prediction of peptide stability, membrane permeability, and pharmacokinetic profiles. By synergizing these advancements, next-generation oral peptides hold transformative potential in precision medicine, enabling targeted therapies for oncology, antimicrobial-resistant infections, and metabolic disorders, ultimately bridging the gap between preclinical promise and real-world therapeutic impact.

## Figures and Tables

**Figure 1 ijms-26-05131-f001:**
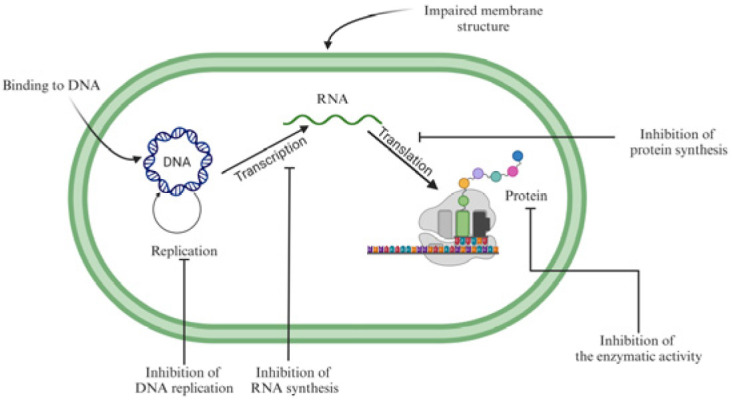
The mechanism of AMPs.

**Figure 2 ijms-26-05131-f002:**
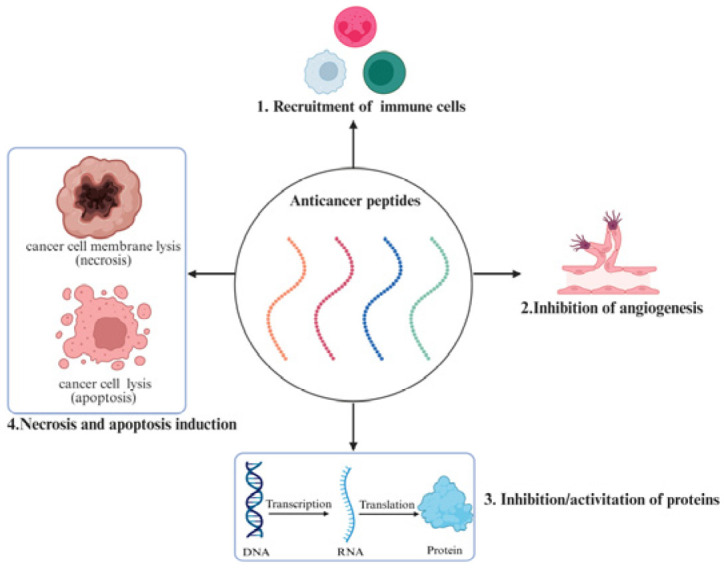
The mechanism of ACPs.

**Figure 3 ijms-26-05131-f003:**
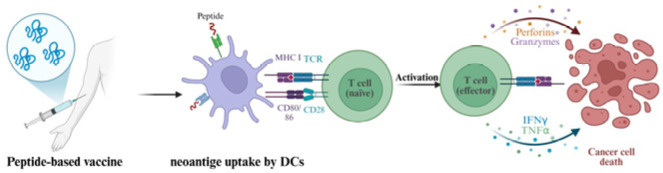
The mechanism of peptide-based vaccines.

**Table 1 ijms-26-05131-t001:** Targets and indications of peptide drugs recently approved by the FDA.

Name of Peptide Drug	Target	Indications
Yorvipath	Parathyroid hormone receptor	Hypoparathyroidism in adults
Trofinetide	Insulin-like growth factor 1 (IGF-1)	Rett syndrome in patients aged ≥2 years
Rezafungin	β-1,3-glucan synthase inhibitor	Adult patients with candidemia and invasive candidiasis
Flotufolastat F18	Prostate-specific membrane antigen (PSMA)	Metastatic prostate cancer
Motixafortide	CXCR4 antagonist	Stem cell mobilization for autologous transplantation in multiple myeloma
Zilucoplan	Complement C5 inhibitor	Generalized myasthenia gravis (anti-AChR antibody-positive adults)
Tirzepatide	GIP and GLP-1 receptors	T2DM and Obesity
Terlipressin	V1 and V2 receptors	Hepatorenal syndrome with rapid reduction in kidney function
Vosoritide	Natriuretic peptide receptor B (NPR-B)	Achondroplasia
Melphalan flufenamide	Exerts anti-tumor activity through cross-linking of DNA	Multiple myeloma (MM) and amyloid light-chain amyloidosis
Voclosporin	T-cells	Lupus nephritis
Pegcetacoplan	Complement protein C3 and its activation product C3b	Paroxysmal nocturnal hemoglobinuria

**Table 2 ijms-26-05131-t002:** Comparative analysis of chemically synthesized and biologically synthesized peptides.

Parameters	Chemical Synthesis	Biological Synthesis
Peptide Length	Optimal for short peptides (≤50 amino acids)	Suitable for long peptides(>50 amino acids)
Cost Efficiency	High reagent costs; limit large-scale feasibility; poor scalability	Cost-effective at scale; highly scalable via fermentation/bioreactor systems
Purity and Impurities	Truncated sequences, racemization, and side reactions necessitate HPLC purification	Host-cell proteins, endotoxins, and misfolding require stringent downstream purification
Modification Flexibility	Enables non-natural amino acids, isotopic labeling, and site-specific modifications	Limited to natural amino acids; modifications demand genetic/enzymatic engineering
Production Time	Shorter cycles (days to weeks) for simple peptides	Extended timelines (weeks to months) due to biological system complexity
Batch Consistency	High reproducibility for short sequences; variability escalates with length/complexity	Potential batch variability from biological instability (e.g., mutations, expression drift)
Environmental Impact	Solvent-intensive processes generate hazardous waste	Reduced solvent use; more sustainable for large-scale manufacturing

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
