# Peer review of "Therapeutic Peptides: Recent Advances in Discovery, Synthesis, and Clinical Translation"

_ijms, 2025, doi:10.3390/ijms26115131_

Round 1

Reviewer 1 Report

Comments and Suggestions for Authors

This article provides a comprehensive review of research advances in therapeutic peptides, systematically summarizes discovery strategies, synthesis technologies, and clinical translation/applications, and integrates emerging technologies like artificial intelligence and synthetic biology. It holds significant reference value for field development. The paper is well-structured with substantial content, particularly demonstrating strong clinical relevance in the applications section (e.g., antimicrobial peptides, anticancer peptides, and vaccines). This is a high-quality domain review recommended for acceptance after minor revisions.
Specific Issues Requiring Attention:

1. "7.Patents" section: The authors did not discuss patent-related content, yet retained this subsection heading. Please verify whether this is an error.
2. Reference formatting issues: For example, reference 24 lacks page number information:
"24. Ahmed I, Asgher M, Sher F, Hussain S, Nazish N, Joshi N, et al. Exploring Marine as a Rich Source of Bioactive Peptides: Challenges and Opportunities from Marine Pharmacology. Marine Drugs. 2022;20(3)."
(Should follow journal format requirements)
3. Content supplementation:
3.1 Need to discuss challenges in peptide vaccine development (e.g., HLA restriction) and corresponding overcoming strategies
3.2 Should highlight unresolved issues (e.g., oral peptide delivery) in the "Prospects" section
3.3 Suggested to supplement current status and bottlenecks of oral peptide drug development in the "Prospects" section (e.g., intestinal permeability, enzymatic stability)

Author Response

Comment1: "7.Patents" section: The authors did not discuss patent-related content, yet retained this subsection heading. Please verify whether this is an error.
Response1: Agree. We have, accordingly, verified the error of the “7、Patents” subsection heading and have deleted it.

Comment2: Reference formatting issues: For example, reference 24 lacks page number information.
Response2: Agree. We have, accordingly, modified the reference formatting based on journal requirements.

Comment3: Need to discuss challenges in peptide vaccine development (e.g., HLA restriction) and corresponding overcoming strategies.
Response3: Agree. We have, accordingly, discussed the challenges and overcoming strategies to emphasized this point, including a) Multi-epitope vaccines Vs HLA restriction, b)Neoantigen vaccines Vs tumor heterogeneity, and c) Adjuvant systems Vs variable immunogenicity. Mention exactly where in the revised manuscript this change can be found—5.3. Peptide-bases Vaccines, Paragraph 3, Page 11 and 421-436 lines.

Comment4: Should highlight unresolved issues (e.g., oral peptide delivery) in the "Prospects" section and Suggested to supplement current status and bottlenecks of oral peptide drug development in the "Prospects" section (e.g., intestinal permeability, enzymatic stability).
Response4: Agree. We have, accordingly, discussed critical barriers of oral peptide drug and future innovations to emphasized this point in “Prosepects” section. Mention exactly where in the revised manuscript this change can be found—6. Conclusion and Prospects, Paragraph2 , Page 12 and 477-493 lines.

Reviewer 2 Report

Comments and Suggestions for Authors

The manuscript titled ‘Therapeutic Peptides: Recent Advances in Discovery, Synthesis, and Clinical
Translation’ of Zheng et al. reports a review of the state-of-the-art in therapeutic peptides. The
authors describe diverse aspects of peptide medicinal chemistry ranging from discovery platforms, synthetic strategies, clinical applications and report some of the most important peptide drugs discovered in the past or present in  recent pipeline. 

The manuscript can be accepted after major revision.

Major Issues
- A prior review by Rossino et al., published on Molecules in 2023 titled "Peptides as
Therapeutic Agents: Challenges and Opportunities in the Green Transition" covers
partially overlapping material. While the thematic focus of the present submission is
different (emphasizing clinical translation), the Introduction section in particular exhibits
notable similarities in content, order, and structure with the earlier work. While no direct
textual duplication is present, the use of similar sentence sequencing, conceptual flow, and
reworded phrasing may represent some concerns.

The authors should substantially revise the Introduction to ensure originality in narrative and
phrasing. Moreover, and more importantly, of reviews on the same topics should be included, clarifying the scope and added value of this new review.

- More importantly, while the review is informative, much of the discussion is too descriptive, it is not a critical review. There is limited critical analysis or a degree of details
reported. A stronger emphasis and analytical discussion would enhance the scientific value.
Consider adding a perspective in the conclusion section, outlining future directions in
peptide drug development.

Author Response

Comment1: A prior review by Rossino et al., published on Molecules in 2023 titled "Peptides as Therapeutic Agents: Challenges and Opportunities in the Green Transition" covers partially overlapping material. The authors should substantially revise the Introduction to ensure originality in narrative and phrasing.
Response1: Agree. We have, accordingly, restructured the Introduction section through a problem-oriented narrative, integrating technology clusters and translational medicine perspectives. The revised framework progresses systematically from technological breakthroughs to application scenarios and ultimately clinical translation,  diverging from conventional  frameworks that progress from historical context through technical developments to market analyses in prior reviews. 

Comment2: while the review is informative, much of the discussion is too descriptive.there is limited critical analysis or a degree of details.
Response2: Agree. We have, accordingly, revised the critical analysis of emerging technologies in peptide drug development to rigorously address their inherent limitations. For instance: 
a.Computer-aided peptide drug design was scrutinized for its heavy reliance on data-dependent algorithms and insufficient incorporation of pharmacodynamic parameters (Revised manuscript can be found—3.4. Computer-Aided Drug Design(CADD) for Peptide Drug Discovery,Paragraph2 , Page 5 and 156-1633 lines.)
b.Recombinant DNA technology was critically evaluated for the limitations of prokaryotic expression systems (e.g., inefficiency in complex disulfide bond networks or glycosylation). (Revised manuscript can be found—4.2.1. Recombinant DNA Technology: Precision Engineering for complex peptides,Paragraph2 , Page 7 and 229-232 & 236-240 lines.)
c.Enzymatic synthesis was analyzed for its dependency on costly cofactors (e.g., ATP, NADPH) and substrate specificity constraints, which hinder scalability and cost-effectiveness. (Revised manuscript can be found—4.2.2. Enzymatic Synthesis: Catalytic Precision for Tailored Therapeutics , Page 7 and 248-258 lines.)
d. Microbial cell factories were assessed for persistent challenges such as host-cell protein (HCP) contamination during large-scale fermentation, necessitating stringent downstream purification protocols. (Revised manuscript can be found—4.2.3. Microbial Cell Factories: Sustainable Planfroms for Industrial-Scale Production , Page 7 and 270-277 lines.)
Also , we discuss barriers of peptide drug development , such as oral administration. (Revised manuscript can be found—6. Conclusion and Prospects , Page12 and 477-484 lines.)

Comment3: Consider adding a perspective in the conclusion section, outlining future directions in peptide drug development.
Response3: Agree. We have, accordingly, modified the unsolved issues for peptide drug development and future innovations to overcome challenges to to emphasized this point. Mention exactly where in the revised manuscript this change can be found—6. Conclusion and Prospects, Paragraph2 , Page 12 and 484-493 lines.